# Analysis of Trace Heavy Metal in Solution Using Liquid Cathode Glow Discharge Spectroscopy

**DOI:** 10.3390/s24237756

**Published:** 2024-12-04

**Authors:** Duixiong Sun, Xinrong Ma, Jiawei Chang, Guoding Zhang, Maogen Su, Marek Sikorski, Vincent Detalle, Xueshi Bai

**Affiliations:** 1Key Laboratory of Atomic and Molecular Physics & Functional Materials of Gansu Province, College of Physics and Electronic Engineering, Northwest Normal University, Lanzhou 730070, China; sundx@nwnu.edu.cn (D.S.); maxinrong2021@163.com (X.M.); changjw1118@163.com (J.C.); zhanggd@nwnu.edu.cn (G.Z.); sumg@nwnu.edu.cn (M.S.); 2Faculty of Chemistry, Adam Mickiewicz University, 61-614 Poznań, Poland; marek.sikorski@amu.edu.pl; 3Centre de Recherche et de Restauration des Musées de France (C2RMF), 75008 Paris, France; 4SATIE, Systèmes et Applications des Technologies de l’Information et de l’Energie, CY Cergy-Paris Université, ENS Paris-Saclay, CNRS UMR 8029, 5 mail Gay Lussac, 95031 Neuville sur Oise, France; vincent.detalle@cyu.fr; 5Fondation des Sciences du Patrimoine/EUR-17-EURE-0021, 95000 Cergy-Pontoise cedex, France

**Keywords:** atmospheric pressure glow discharge, atomic emission spectroscopy, trace heavy metals, detection limit, sewage

## Abstract

Heavy metal pollution, particularly from cadmium (Cd) and copper (Cu), poses significant environmental and health risks. To address the need for efficient, portable, and sensitive detection methods, this study introduces an improved atmospheric pressure glow discharge atomic emission spectrometry (APGD-AES) technique for quantifying Cd and Cu in water samples. The APGD-AES method offers key advantages, including low energy consumption (<33 W), high excitation energy, and compact design. The system was optimized for a discharge voltage of 550 V (Cd) and 570 V (Cu), a flow rate of 3.6 mL/min, and a solution pH of 1.0. Under these conditions, detection limits reached 16 µg/L for Cd and 1.3 µg/L for Cu. APGD-AES was tested on real water samples, including sewage and tap water, demonstrating compliance with national safety standards and comparable performance to graphite furnace atomic absorption spectrometry (GFAAS). This technique shows promise for real-time, on-site monitoring of trace heavy metals due to its portability, precision, and cost-efficiency.

## 1. Introduction

Human activities and industrial development contribute to toxic element pollution, a significant environmental issue that threatens human health when element concentrations exceed safe limits. The determination of toxic elements like cadmium (Cd) and copper (Cu), which are bioaccumulative, is critical [1]. Worldwide, strict regulations have been established for the maximum allowable concentration of toxic heavy metals in the environment [2].

Several methods have been developed for the measurement and monitoring of low-concentration toxic elements, such as atomic absorption spectrometry [3], inductively coupled plasma atomic emission spectrometry (ICP-AES) [4], and inductively coupled plasma mass spectrometry (ICP-MS) [5]. However, the application of these methods in the laboratory is restricted, and usually necessities complex equipment and operations due to the instrument size, power consumption, and requirement for a vacuum environment [6,7,8]. So, these inconvenient conditions limit their use for rapid analysis under field conditions.

To address the need for miniaturized and portable online detection, Cservalvi et al. developed the electrolyte cathode atmospheric glow discharge (ELCAD) device, demonstrating its potential as a unique emission source for continuous multi-metal detection [9,10]. In recent years, the ELCAD device and similar devices have been widely used in analysis of various complex samples such as soil and spruce needles; colloidal silica [11]; zirconium alloys [12]; human hair [13]; honey [14]; and water with varying properties (from lakes, rivers, taps, or minerals) [15], including high-salinity waters such as seawater, wastewater [16], saline water [17], and blood samples [18].

Since the ELCAD device was first constructed in 1993, a series of optimizations have been implemented on the experimental device to enhance discharge stability and portability: liquid sampling–atmospheric pressure glow discharge [19], solution cathode glow discharge [20,21], DC atmospheric pressure glow discharge [22,23], AC electrolyte atmospheric liquid discharge (AC-EALD) [24,25], hanging droplet cathode atmospheric pressure glow discharge (HDCAPGD) [26], and hollow anode–liquid cathode glow discharge (HA-SCGD) [27]. Optimized ELCAD systems can achieve analysis performance comparable to or surpassing ICP-AES in laboratories. However, practical applications face challenges, such as replacing costly platinum electrodes with cheaper, more stable alternatives and addressing liquid-surface fluctuations to maintain a consistent distance between the liquid surface and electrode for stable discharge.

In this work, we developed a flowing liquid cathode atmospheric pressure glow discharge atomic emission spectroscopy technique for determination of water samples. The improved ELCAD device demonstrated multiple advantages over traditional systems. A needle-shaped tungsten anode enhanced discharge efficiency while reducing energy consumption (<33 W) and costs. Encasing the capillary quartz tube within graphite rods streamlined the structure, minimized sample loss, and increased the solution-to-rod contact area, boosting conductivity. Plasma stability was achieved through peristaltic pump modifications.

The levels of Cd and Cu in the liquid were identified to evaluate the feasibility of this method, and the effects of key parameters such as discharge voltage, solution pH, and solution flow rate on emission intensity were comprehensively investigated. The detection limits, accuracy, and precision of Cu and Cd measurements obtained in this study were compared with those reported for other ELACD techniques.

Access to safe drinking water is a fundamental requirement for human survival. However, with the rapid economic development in recent times, the increasing discharge of industrial wastewater has led to higher concentrations of heavy metals in water, significantly affecting water quality and posing risks to human health. This technology (APGD-AES) offers advantages such as portability, high precision, short detection cycles, and low cost, making it a promising and efficient method for water quality monitoring.

## 2. Experimental

### 2.1. Reagents and Samples

The stock solution, containing concentrations of Cd and Cu at 1000 mg/L, was purchased from the Beijing Institute of Metrology and Quality Inspection-Standard Material Center (China). Before use in the measurement process, the solution had to be diluted. Deionized water was used in the dilution process to ensure the stability of the metal content being tested. The actual water samples used in the experiment were collected from the tap water of the Physics and Electronic Engineering College of Northwest Normal University, as well as from the sewage discharged by the Annin District Sewage Treatment Plant in Lanzhou City, Gansu Province, China.

Guaranteed reagent (GR) grade nitric acid (HNO_3_) was purchased from Ningbo Guoyao Chemical Reagent Co., Ltd. (Ningbo, China). Nitric acid (HNO_3_) was used to adjust the pH of the solution, and the pH value of the solution was measured using a PHS-3E pH meter (INESA, Shanghai, China).

### 2.2. Instrumentation

Glow discharge represents one of the fundamental forms of electric breakdown discharge. When two electrodes are subjected to a gradually increasing DC voltage, an electric field is induced. At this time, free electrons are accelerated by the electric field and acquire kinetic energy. These energetic electrons collide with other particles, leading to avalanche ionization, and consequently, to an exponential increase in the number of free electrons. As the count of electrons and ions introduced to the plasma is equal to the number of ions and electrons dissipating within the plasma, the discharge becomes self-sustaining, which causes the onset of the glow discharge stage.

The experimental setup device shown in Figure 1 demonstrates the transportation of the sample solution via a peristaltic pump at a flow rate of 3.6 mL/min. To reduce fluctuations of the liquid surface caused by the peristaltic pump on the sample solution, several knots (acting as dampers) were tied at one end of the peristaltic pump’s rubber tube. Additionally, a Murphy’s dropper was connected to the other end to diminish the fluctuation impact and enhance the stability of the plasma.

The excitation system comprised two primary components: a tungsten wire (with diameter 0.5 mm and length 3 mm) serving as the anode, and the sample solution acting as the cathode. The conductive nature of the sample solution resulted from its overflows from the top of the J-shaped capillary, which had an inner diameter of 1.0 mm and an outer diameter of 1.2 mm. Marinating a gap of 2.0 mm between the needle-shaped tungsten electrode and the quartz capillary, the capillary was encircled by four graphite rods with a diameter of 5 mm. A DH1722A-6-type DC power (Beijing Dahua Radio Instruments Co., Ltd., Beijing, China) supply was used to provide voltage. A liquid overflow from the capillary’s top, positioned 2 mm above the graphite rod, ultimately entered the waste liquid tank.

In an ambient air environment, high voltage was applied between the two electrodes, exciting the solution through high-energy electrons to generate a stable glow discharge. The glow was directed onto the spectrometer equipped with a charge-coupled device detector preceded by an image intensifier (iStar-DH734-18F-03, Andor Technology, Belfast, Northern Ireland, UK) through a quartz convex lens (ƒ = 100 mm), and the plasma emission signal was detected using Andor software version 4.27.30001.0. The optical resolution of the spectrometer was around 0.05 nm with grating of 2400 lines/mm.

## 3. Results and Discussion

### 3.1. Spectral Characteristics of Reagents and Solutions of APGD-AES

The reagents used to prepare the solution must be free of the target element to prevent interference with its content. While we had selected high-purity reagents for the experiment, it was still crucial to verify their purity prior to proceeding. Figure 2 shows the emission spectrum of a blank HNO_3_ solution with a pH of 1.0 within the wavelength range of 270–700 nm. The emission lines observed around 310 nm are attributed to the molecular band of OH (A^2^∑^+^ (ν′ = 0)→X^2^Π (ν″ = 0)) [28] radical, resulting from the ongoing evaporation and vaporization of water molecules during the discharge process. During this process, electrons interact with water molecules, ultimately producing OH. The wavelength range of 315.0–435.0 nm represents the molecular band of N_2_ (SPS, C3Πu→B3Πg), and includes five vibrational sequences, i.e., Δν = 0, −1, −2, −3, and −4, and a total of 20 lines, originating from the fact that the experiments were conducted in an air environment. Within the 390–435 nm wavelength range, the N_2_ band overlaps extensively with the spectral lines of O II line produced by water vapor. The emission lines at 656.27 nm and 486.12 nm correspond to the atomic lines of Hα and Hβ, respectively [29]. Additionally, the presence of the wavelength 589.12 nm indicates the characteristic line of Na I, suggesting the existence of some impurities within the blank solution.

To determine the characteristic spectral lines of the element to be analyzed, 3 mg/L Cu solutions and 1 mg/L Cd solutions, both with a pH of 1.0, were used to conduct APGD-AES experiments. The emission spectra of these solutions were compared to that of the blank solution. Figure 3 illustrates the emission spectra of the 3 mg/L Cu solution and the 1 mg/L Cd solution, both with a pH of 1.0. Notably, in addition to the emission spectral lines observed in the blank solution, new spectral lines emerged at 324.86 nm and 327.49 nm, which correspond to the emission spectral lines of Cu I. Due to its high intensity and lack of interference, the spectral line at 324.86 nm was selected as the analytical line for Cu I. In Figure 3b, the atomic line of Cd appears at 228.87 nm. This isolated spectral line remains free from interference by other emission spectral lines. Consequently, 228.87 nm was chosen as the analytical line for Cd I. These results indicate that the experimental reagents did not interfere with the target element during the solution preparation process.

### 3.2. Optimization of Operating Parameters

#### 3.2.1. Effect of Discharge Voltage on Emission Signals

Discharge voltage plays a crucial role in elemental emission, as it directly influences the excitation energy. Preliminary experiments revealed that discharge voltages below 500 V resulted in weak plasma formation, with most energy dissipated in heating and evaporating water, leading to undetectable emission lines of Cu and Cd. Conversely, voltages exceeding 580 V caused the tungsten wire to glow red and resulted in solution spillage from the capillary. Exceeding 650 V further exacerbated this issue, leading to the melting of the tungsten wire and capillary. To optimize emission signals from Cd and Cu, we investigated the effects of varying discharge voltages within the range of 510 V to 580 V.

As illustrated in Figure 4, increasing the discharge voltage led to a corresponding increase in the emission intensity of Cd and Cu. This enhancement was attributed to the increased excitation energy, resulting in a higher density of excited metal atoms and improved excitation efficiency.

To optimize the signal-to-background ratio (SBR) and signal intensity, we determined the optimal discharge voltages for Cd and Cu. For Cd, a discharge voltage of 550 V provided the best SBR and moderate signal intensity. Similarly, for Cu, 570 V was found to be the optimal discharge voltage based on SBR and signal intensity considerations. The SBR of the spectral lines in Figure 4 shows a slight decreasing trend after the discharge voltage reached 570 V. This was attributed to the fact that, with the increase in discharge voltage, not only did the intensity of the spectral lines of interest increase, but the background continuous signal from blackbody radiation also increased, leading to a decreasing trend in the SBR.

#### 3.2.2. Effect of Solution Flow Rate on Emission Intensity

The stability of glow discharge is intimately linked to the liquid cathode’s flow rate. Insufficient flow rate leads to a rapid decrease in liquid level due to thermal evaporation, resulting in a highly unstable plasma. Conversely, excessive flow rate can induce rapid quenching of the plasma through interaction with the cold liquid. Therefore, optimizing the liquid flow rate is crucial for maintaining a stable discharge process. During this experiment, a weak and unstable glow was observed when the flow rate was maintained below 3.0 mL/min. On the other hand, flow rates exceeding 6.0 mL/min resulted in significant sample consumption and splashing. Consequently, we investigated the impact of flow rates between 3.0 and 6.0 mL/min on the emission intensity of Cd and Cu.

As show in Figure 5, the emission intensity of Cd and Cu varied with the flow rate. For Cu, increasing the flow rate from 3.0 to 3.6 mL/min led to an increase in emission intensity. This was attributed to the increased number of analyte atoms entering the discharge zone. However, further increasing the flow rate beyond 3.6 mL/min resulted in a decrease in Cu emission intensity due to increased water loading, which reduced the energy available for sample excitation. For Cd, as the flow rate increased, the spectral line intensity generally showed an upward trend within the margin of error; this was also attributed to the increase in the number of analyte atoms entering the discharge region. The highest signal intensity was observed at a flow rate of 5.4 mL/min, although plasma stability was compromised. Upon calculating the relative standard deviation (RSD), a flow rate of 3.6 mL/min was found to provide optimal plasma stability for both Cu and Cd, with RSD values of 2.52% and 1.30%, respectively.

#### 3.2.3. Effect of Solution pH on Emission Signal Intensity

It was observed that at pH values below 0.9, the conductivity sharply increased, causing the W needle to become red-hot due to high temperatures. Simultaneously, the solution surrounding the anode rapidly boiled, leading to poor plasma stability. Conversely, at pH values exceeding 1.3, conductivity decreased, resulting in diminished emission signal intensity, a weaker glow, and a smaller excitation-generated plasma. Therefore, the investigation focused on studying the variation in emission intensity within the pH range of 0.9–1.3.

As illustrated in Figure 6, the emission signal intensity decreased with increasing pH. This decline was attributed to decreased solution conductivity, because the conductivity of a solution depends on the concentration of H⁺ ions in the solution. As the pH increases, the concentration of H⁺ ions decreases, resulting in a decrease in the solution’s conductivity, which requires more energy for heating and evaporation of the solution, consequently inhibiting element excitation. At pH 1.0, a moderate intensity with a low coefficient of variation (Table 1) and good stability was observed. Hence, pH 1.0 was selected as the optimal parameter.

### 3.3. Analytical Performance

The analytical performance of APGD-AES for Cd and Cu was evaluated under optimal experimental conditions. Calibration curves for Cd and Cu were constructed by linearly fitting the characteristic emission line intensity to the solution concentration, as shown in Figure 7. The emission intensities of Cd and Cu exhibited a linear relationship within the concentration range of 30–100 µg/L, with the relevant parameters summarized in Table 1.

The limit of detection (LOD) is calculated by LOD = kσ/S [30], where k represents the confidence coefficient, typically set as k = 3; σ denotes the standard deviation of the blank signal; and S signifies the slope of the calibration curve. Consequently, the detection limits of Cd and Cu were determined as 16 µg/L and 1.3 µg/L, respectively. In order to estimate the power consumption of the system during the measurement process, the volt–ampere characteristic curves of Cd and Cu within the 470–600 V range were established, as shown in Figure 8. It is evident that the discharge power for Cd and Cu was measured at 32 W and 23 W, respectively, indicating that minimal power consumption was required for the determination of these elements. The variance in discharge power under identical conditions was attributed to the distinct conductivities of Cd and Cu.

Table 2 compares the detection limits of Cd and Cu measured by this method with those obtained using other ELCAD-AES devices. It is evident that the detection limits achieved by this method were superior to those reported for most other devices. This improvement can be attributed to the systematic optimization of our system and the use of an ICCD detector. The results demonstrate that APGD-AES offers high sensitivity, low detection limits, low power consumption, stability, and simplicity, making it a promising analytical technique. Experimental results show that the detection limit for Cd was higher than for Cu. This is attributed to Cd’s higher excitation energy compared to Cu, which results in weaker emission lines from the Cd plasma, making its signal less intense than that of Cu.

### 3.4. Detection of Cu and Cd Elements in Real Water Samples

Based on the above experimental conditions, this method was employed to analyze the sewage of the Anning sewage treatment plant in Lanzhou, Gansu Province, and tap water of the Anning District. The obtained results were compared with those from graphite furnace atomic absorption spectrometry (GFAAS) and benchmarked against both the national sewage discharge standards and drinking water standards. The results are shown in Table 3.

In the sewage sample, Cd was not detected. The national sewage discharge standard stipulates that the Cd concentration should be less than 10 µg/L. Regarding Cu, its concentration measured 10.9 µg/L, which falls below the first-level standard of 0.5 mg/L outlined in the national sewage discharge standard [37]. Moving to tap water analysis, the content of Cu was found to be 1.3 µg/L, which is lower than the hygienic standard for drinking water set at 1 mg/L [38]. These results imply that the technology can be used to monitor excessive heavy metals in drinking water.

The experimental results show that there were differences between atmospheric pressure glow discharge atomic emission spectrometry (APGD-AES) and graphite furnace atomic absorption spectrometry (GFAAS) in the measurement results, which were due to their respective analytical principles and technical characteristics. Although the sample pretreatment in GFAAS ensures the accuracy of the results, it also limits its real-time monitoring efficiency. In contrast, APGD-AES only requires adjusting the pH of the sample and optimizing the equipment to be suitable for industrial real-time detection. Although there were differences in measurement results between APGD-AES and GFAAS, these deviations are acceptable to some extent.

## 4. Conclusions

We have demonstrated in this work that the atmospheric pressure glow discharge atomic emission spectrometry (APGD-AES) technique achieved the quantification of Cd and Cu in a solution by optimizing the optimal experimental conditions as a discharge voltage of 550 V (Cd)/570 V (Cu), a solution flow rate of 3.6 mL/min, and a pH of 1.0. The energy consumption of APGD-AES was found to be less than 33 W, and the detection limits for Cd and Cu were determined to be 16 µg/L and 1.3 µg/L, respectively.

The concentrations of Cu and Cd in both the sewage from the Anning Wastewater Treatment Plant and the tap water in Anning District were measured, and the results were compared with those obtained by GFAAS. The findings indicated that the levels of these metals in both types of water samples conformed to the relevant standards. APGD-AES technology demonstrates the benefits of low energy consumption, cost-effectiveness, and high analytical precision. In addition, the device can be implemented with smaller dimensions then the traditional ELCAD instruments, which offers the great potential to serve as a portable device for real-time, on-site, and continuous monitoring of toxic trace heavy element contents in environmental settings.

## Figures and Tables

**Figure 1 sensors-24-07756-f001:**
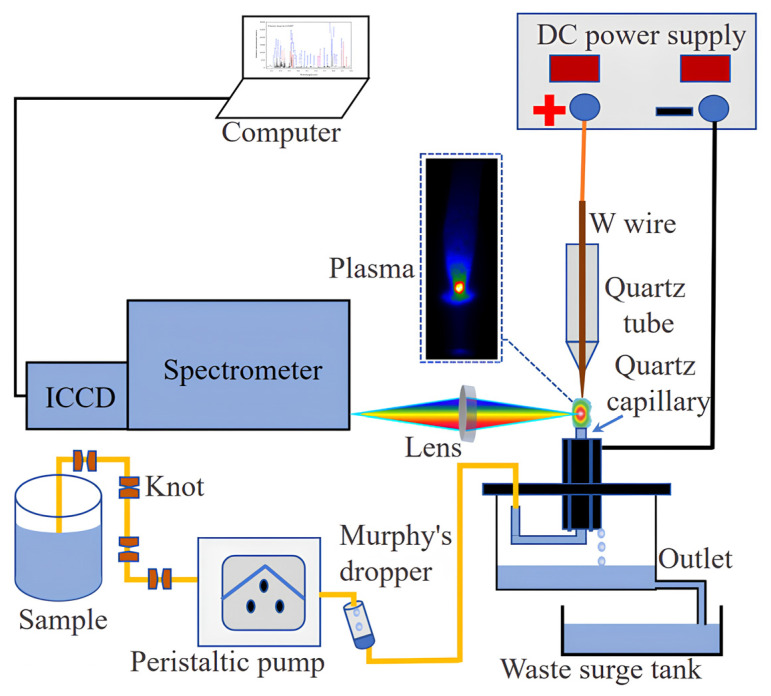
Instrumentation for atmospheric pressure glow discharge atomic emission spectrometry, APGD-AES.

**Figure 2 sensors-24-07756-f002:**
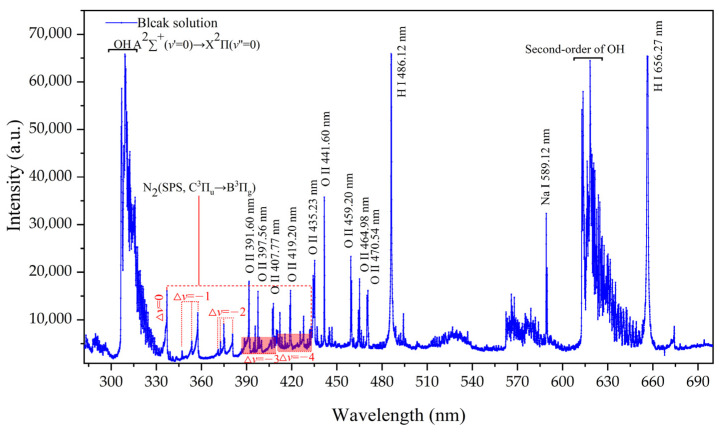
Emission spectrum of blank solution in 270–700 nm range. Electrolyte: HNO_3_, pH = 1; discharge voltage: 550 V; liquid flow rate: 3.6 mL/min; interelectrode gap: 1 mm.

**Figure 3 sensors-24-07756-f003:**
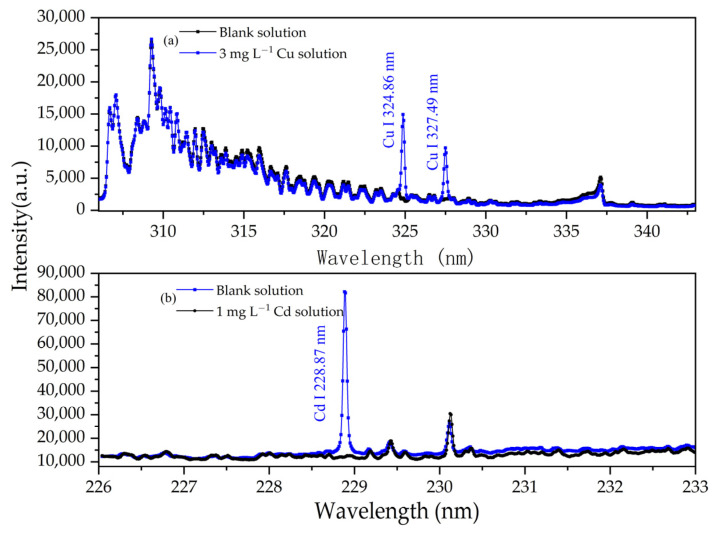
Emission spectra of blank solution and (**a**) 3 mg/L Cu solution and (**b**) 1 mg/L Cd solution. Electrolyte: HNO_3_, pH = 1; discharge voltage: 570 V; liquid flow rate: 3.6 mL/min; interelectrode gap: 1 mm.

**Figure 4 sensors-24-07756-f004:**
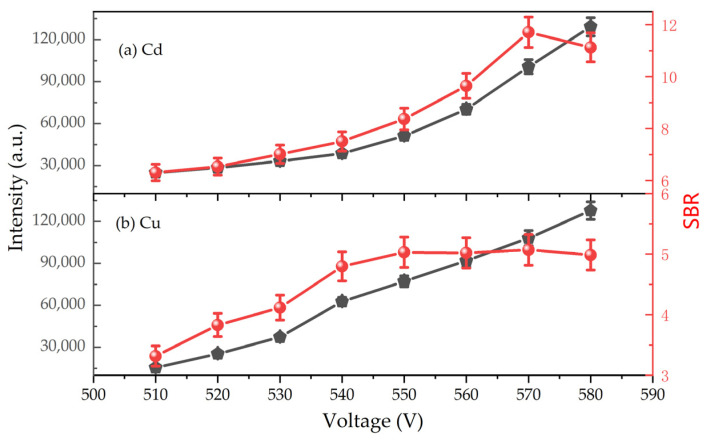
Effect of discharge voltage on the emission intensity and signal-to-background ratio of Cd I and Cu I. Electrolyte: HNO_3_, pH = 1; liquid flow rate: 3.6 mL/min; interelectrode gap: 1 mm; concentration of Cd solution: 1 mg/L; concentration of Cu solution: 3 mg/L.

**Figure 5 sensors-24-07756-f005:**
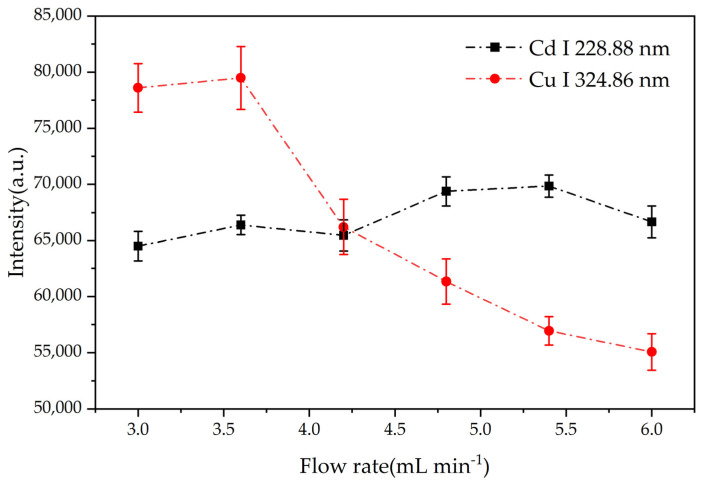
Effect of solution flow rate on emission signal intensity. Electrolyte: HNO_3_, pH = 1; discharge voltage: 550 V (Cd)/570 V (Cu); concentration of Cd solution: 1 mg/L; concentration of Cu solution: 3 mg/L; interelectrode gap: 1 mm. The error bars represent the standard deviation of the spectral intensity after 10 measurements.

**Figure 6 sensors-24-07756-f006:**
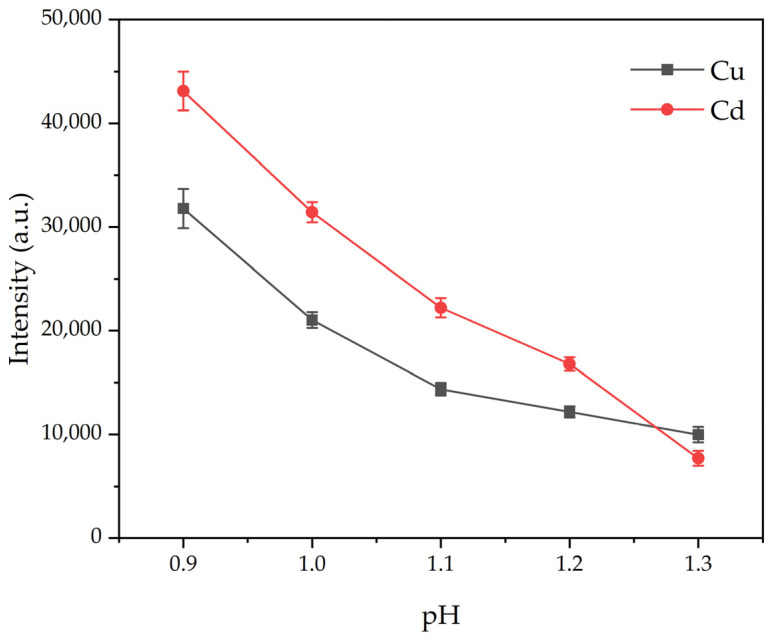
Effect of solution pH on emission signal intensity. Electrolyte: HNO_3_; discharge voltage: 550 V (Cd)/570 V (Cu); liquid flow rate: 3.6 mL/min; concentration of Cd solution: 1 mg/L; concentration of Cu solution: 3 mg/L; interelectrode gap: 1 mm.

**Figure 7 sensors-24-07756-f007:**
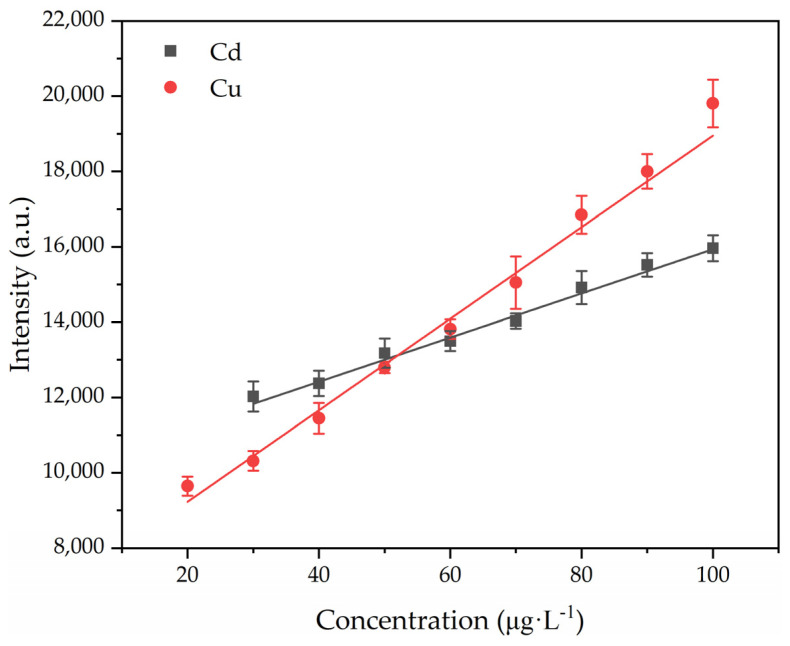
Calibration curves for Cu and Cd. Electrolyte: HNO_3_, pH = 1.0; discharge voltage: 550 V (Cd)/570 V (Cu); liquid flow rate: 3.6 mL/min; interelectrode gap: 1 mm.

**Figure 8 sensors-24-07756-f008:**
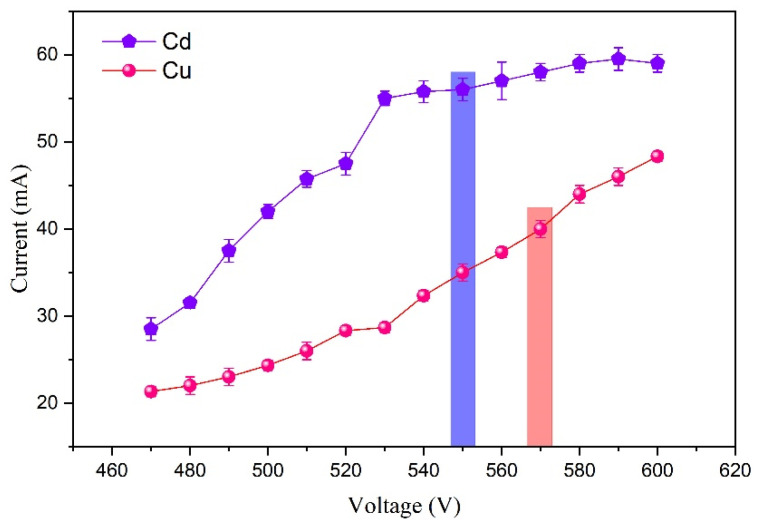
The voltametric characteristic curve of Cu and Cd. Electrolyte: HNO_3_, pH = 1.0; discharge voltage: 550 V (Cd)/570 V (Cu); liquid flow rate: 3.6 mL/min; interelectrode gap: 1 mm.

**Table 1 sensors-24-07756-t001:** Analytical performance using the optimal experimental conditions.

Species	Analytical Line (nm)	Power (W)	Calibration Equation	R^2^	Sensitivity	LOD (μg/L)	RSD ^a^/%
Cd	228.87	30.2–32.4	*I* = 10,077.4 + 58.511*C*	0.9846	58.511	16	2.9
Cu	324.86	22.3–23.3	*I* = 4666.8 + 152.81*C*	0.9979	152.81	1.31	4

^a^ After 10 measurements at a solution concentration of 50 μg/L.

**Table 2 sensors-24-07756-t002:** The detection limits of this method are compared with those of other methods.

Method	LOD (μg/L)		Reference
	Cd	Cu	
Atmospheric pressure glow discharge atomic emission spectrometry	16	1.3	This work
Hollow anode–liquid cathode glow discharge	2	8	[31]
Modified electrolyte cathode atmospheric glow discharge	5	11	[32]
Liquid cathode glow discharge atomic emission spectrometry	370	470	[33]
Liquid sampling atmospheric pressure glow discharge	50	650	[34]
Flowing liquid cathode atmospheric pressure glow discharge optical emission spectroscopy	10	-	[35]
Flowing liquid anode atmospheric pressure glow discharge	6	-	[36]

**Table 3 sensors-24-07756-t003:** Atmospheric pressure glow discharge atomic emission spectrometry (APGD-AES) test results of real water samples.

Sample	Element	Measured Value (µg/L)	GFAAS (µg/L)	The Maximum Value Set by the State (µg/L)
Sewage	Cd	—	0.09	10
	Cu	10.9	9.72	500
Tap water	Cd	—	0.04	5
	Cu	3.10	2.33	1000

## Data Availability

The original contributions presented in this study are included in the article. Further inquiries can be directed to the corresponding author.

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
