# Peer review of "Analysis of Trace Heavy Metal in Solution Using Liquid Cathode Glow Discharge Spectroscopy"

_sensors, 2024, doi:10.3390/s24237756_

Round 1
Reviewer 1 Report
Comments and Suggestions for Authors
This study introduces a low energy consumption and compact design atmospheric pressure glow discharge atomic emission spectroscopy (APGD-AES) method for quantifying Cd and Cu in water samples. And tests were conducted on sewage and tap water to verify the feasibility of the method. Overall, it can be considered a relatively complete study, but there are still some issues that need to be addressed and discussed before publication:
1. The author's language in the research background section is not concise enough, and the introduction of the innovative points of this study is not prominent enough. It is suggested to increase the comparison with existing research and highlight the innovative points of the article
2. The explanation of the principle section in the article is not deep enough. It is suggested to add an explanation of the mechanism to facilitate readers' understanding.
3. Please explain in detail why the red line in Fig.4 shows a downward trend.
4. In section 4.3.4, the measurement values of sewage in the area using other detection methods should be added to verify the accuracy of the method.
Reviewer 2 Report
Comments and Suggestions for Authors
I reviewed the article entitled “Analysis of Trace Heavy Metal in Solution using Liquid Cathode Glow Discharge Spectroscopy” carefully. In this work, the authors report an improved atmospheric pressure glow discharge atomic emission spectrometry (APGD-AES) technique for quantifying Cd and Cu in water samples. In my opinion, this work can be considered for publication in Sensors after some revisions. My suggestions are listed as follows:
(1) At the end of the introduction, industrial implication of this work should be emphasized to substantiate its academic value.(2) For the emission spectrum in Figure 2, each band assignment should be validated by supporting authoritative article.
(3) The limit of detection (LOD) is calculated by LOD=kσ/S: cite reference.
(4) Add error bars for each curve in Figure 4.
(5) As Table 2 lists, detection limit of Cd is much higher than that of Cu. Please elucidate this properly.
(6) In Table 2, method names should appear in full form, rather than abbreviated form.
(7) Explain the fluctuation of Cd signal intensity with flow rate depicted in Figure 5.
(8) This decline is attributed to decreased solution conductivity: why the solution conductivity decreases with pH?
(9) In table, measured values for Cd and Cu with atmospheric pressure glow discharge atomic emission spectrometry deviate from those acquired via Graphite Furnace Atomic Absorption Spectrometry. Please interpret this properly to justify the validity of atmospheric pressure glow discharge atomic emission spectrometry.
(10) Some references should be updated with more recently published ones.
In general, I recommend major revision.
I hope that my comments be constructive to some extent.
Reviewer 3 Report
Comments and Suggestions for Authors
Suna et al. reported an improved atmospheric pressure glow discharge atomic emission spectrometry (APGD-AES) technique for quantifying Cd and Cu in water.
The authors used needle-shaped tungsten instead of platinum as the discharge anode to improve discharge efficiency and reduce the cost and energy consumption (<33W). Moreover, the authors have taken a creative approach to enclose the capillary quartz tube within four graphite rods, resulting in a compact device structure. This innovative approach minimizes sample loss and increases the effective contact area between the solution and the graphite rods, thereby enhancing conductivity.
The authors also systemically studied the effect of discharge voltage on emission signals, the effect of solution flow rate on emission intensity, and the effect of solution pH on emission signal intensity to optimize the performance of the developed device. The device shows optimum performance at a discharge voltage of 550V for Cd and 570V for Cu, at a flow rate of 3.6 mL/min, and a solution pH of 1.0. Under the optimized conditions, the detection limits of the spectrometry reached 16 μg/L for Cd and 1.3 μg/L for Cu. Sewage and tap water samples were also investigated with the developed APGD-AES device, showing comparable performance to conventionally used Graphite Furnace Atomic Absorption Spectrometry (GFAAS).
Overall, the authors have demonstrated a creative approach to developing the APGD-AES device and systemically optimizing its performance. I believe the manuscript is a good fit for the journal.
Round 2
Reviewer 2 Report
Comments and Suggestions for Authors
This work can be considered for publication in its present form.